# OpenReview forum: "Imperceptible Jailbreaking against Large Language Models"
_ICLR.cc/2026/Conference — ICLR 2026 Conference Withdrawn Submission_

### Official Review · Reviewer_747m · 2025-10-26

**Soundness:** 3
**Presentation:** 3
**Contribution:** 2
**Rating:** 2
**Confidence:** 4

**Summary:**

This paper proposes an imperceptible jailbreak technique for LLMs that uses existing Unicode variation selectors to generate malicious prompts so that the on-screen string looks unchanged while tokenization differs. This paper designs a chain-of-search pipeline to maximize the log-likelihood of target-start tokens (e.g., `Sure`) and thereby induce harmful responses. The authors claim high attack success rates across open-sourced models like Vicuna, Llama, and Mistral.
However, the current version has problems in terms of threat model, experimental metrics, and generalization.

**Strengths:**

1. **Clear presentation for core mechanism**. Using the variation selectors as invisible, copy-and-paste-stable codepoints to design invisible attacks.
2. **Large-scale experiments on four models**. Comparing five baseline methods on four models and achieving the highest ASR.

**Weaknesses:**

1. **Unclear attack scenarios and threat model**: The paper lacks a detailed description of the attack scenario and threat model. One of my concerns is that the invisible jailbreak attack (the key contribution highlighted in the title and abstract) has a very narrow application scenario. In what scenarios does a jailbreak attack need to be invisible? Jailbreak attack samples are often directly constructed by the attacker to bypass the safety alignment of LLMs, enabling them to output harmful contents that violate restrictions (such as advocating violence, creating dangerous objects, etc.). This scenario does not require the attack to be invisible because the attack samples do not need to be hidden from the model user (i.e., the attacker himself). Furthermore, if the attack scenario assumes that the attacker constructs the attack sample to demonstrate that the system is not well-aligned, it thereby harms the reputation of the model developer. At this time, constructing a GCG sample in this scenario can also achieve the same purpose. I admit that invisibility is an interesting feature, but in what scenarios does an invisible jailbreak attack really need to be necessary? Perhaps invisible attacks are more suitable for prompt injection attacks, where attackers can use these tokens to guide the model to perform user-desired tasks without the user's knowledge.
2. **Unreliable attack success assessment method**: This paper uses keywords (e.g., "sure") to evaluate attack success, an unreliable method that can easily overstate the effectiveness of the attack. Specifically, the attack is performed by inserting unseen tokens into the original input, without sufficient evidence to demonstrate that these tokens do not alter the semantics of the original question. Therefore, relying on the first token to assess attack success can lead to false positives (e.g., the model may output answers to different questions when facing the attack sample). The authors should consider using advanced judge methods (e.g., using an LLM as a judge) to evaluate the results to support the effectiveness of the attack. Furthermore, the authors should present experimental results (e.g., by providing the model's outputs for the attack instructions) to help reviewers understand the effectiveness of the paper.
3. **Lack of novelty**: Invisible attacks with unicode are well-established [1][2]. The novelty of this paper mainly lies in the chain of search methods. However, I have concerns about the efficiency and generalizability of the search chain method. The authors could report the cost of searching for attack samples and compare it with other jailbreak attack methods, such as GCG (or at least provide additional discussion). In addition, how about the generalizability of this method on mainstream commercial models like GPT-4 (which is mentioned in Table 1)?

**Questions:**

1. What is the attack scenarios and threat model of this paper?  In what scenarios does a jailbreak attack need to be invisible?
2. What are the results of using other advanced attack assessment methods?
3. What is the cost of this attack method?
4. What is the attack effect on mainstream commercial models like GPT-4? There are too many ways to effectively bypass the safety alignment of open source models (e.g., fine-tuning with several samples [3]). The author should try the effect of this attack on commercial models such as GPT-4 to intuitively illustrate the harm and value of this attack method. (Even white-box methods like GCG can explore the effect of transfer attacks on the commercial models)


[1] Bad characters: Imperceptible nlp attacks. S&P, 2022.

[2] Trojan source: Invisible vulnerabilities. USENIX, 2023.

[3] Fine-tuning aligned language models compromises safety, even when users do not intend to! ICLR, 2024

---

### Official Review · Reviewer_MTPL · 2025-10-27

**Soundness:** 3
**Presentation:** 4
**Contribution:** 3
**Rating:** 4
**Confidence:** 4

**Summary:**

This paper introduces imperceptible jailbreaks that leverage Unicode variation selectors to compromise large language models (LLMs). Unlike previous works that rely on visible modifications to prompts, the perturbations in this paper are invisible to human perception. Furthermore, the authors propose a chain-of-search pipeline that optimizes these invisible suffixes to maximize the likelihood of target-start tokens (e.g., “Sure”), thereby steering aligned models to produce harmful responses.

**Strengths:**

1. The exploration of variation selectors as invisible suffixes for jailbreaking LLMs is interesting, and contextually well-motivated.
2. The writing is clear and well-structured — the authors present their ideas step by step, and the methods are clearly defined.
3. The presentation quality is strong. Figures and tables are well-designed and effectively support the main claims.
4. The authors evaluate their approach across multiple LLMs, which enhances the credibility and generalizability of the proposed method.

**Weaknesses:**

1. **Limited datasets.** The paper only conducts experiments on 50 samples from AdvBench, while the full benchmark contains 500 instances. Using merely 10% of the available data substantially limits the validity of the conclusions. A broader evaluation would strengthen the empirical claims.
2. **Insufficient cross-dataset validation.** Although the authors claim to follow the 50-sample evaluation protocol from [1], this paper also involves HarmBench. I strongly recommend evaluating the proposed method on more datasets for better generalization evidence.
3. **Lack of robustness testing.** Figure 1 shows that LLMs are not robust to the proposed variation-selector attack. However, the paper focuses heavily on quantitative success rates without testing any defense mechanisms. It would be better to evaluate the attack under common defense settings (e.g., prompt-based defenses [2], or smoothing-based methods such as SmoothLLM [3]) to assess robustness.
4. **Incomplete analysis of “invisibility.”** Although the perturbations are visually imperceptible after rendering, they remain detectable through string-level inspection (for example, the prompt length). This differs from “invisibility” in image perturbations within multimodal models. The authors should explicitly discuss this limitation.
5. **Unclear computational efficiency.** The paper does not report the runtime or computational cost of the chain-of-search process compared to baseline attacks.

[1] Improved Techniques for Optimization-Based Jailbreaking on Large Language Models

[2] Intention analysis prompting makes large language models a good jailbreak defender.

[3] Smoothllm: Defending large language models against jailbreaking attacks

**Questions:**

Please follow the weaknesses.

I am happy to increase my score if my main concerns are addressed (e.g., broader datasets and computational efficiency presentation).

---

### Official Review · Reviewer_U2fz · 2025-10-31

**Soundness:** 3
**Presentation:** 3
**Contribution:** 2
**Rating:** 2
**Confidence:** 3

**Summary:**

The paper introduces a novel class of "imperceptible jailbreaks" targeting LLMs by exploiting invisible Unicode characters known as variation selectors. Unlike existing jailbreaking techniques, which modify prompts visibly to induce harmful responses, this method employs invisible suffixes appended to harmful prompts. These invisible characters remain undetectable to humans but alter the tokenization process, enabling adversarial manipulation of the model’s output. The paper proposes a chain-of-search optimization pipeline to generate these adversarial suffixes and demonstrate high success rates in bypassing safety alignment in LLMs.

**Strengths:**

● The paper's core contribution is the introduction of imperceptible jailbreaks using invisible variation selectors. This technique offers a novel attack vector, bypassing safety mechanisms without visible change to the prompt.
● The method proves highly effective, achieving high attack success rates against a diverse range of LLMs.
● The paper clearly details the "chain-of-search" pipeline used to optimize the adversarial suffixes, providing a robust methodology that ensures the attack's reproducibility.

**Weaknesses:**

1. While I respect the idea and discovery in this paper, particularly how invisible Unicode characters can induce harmful responses in LLMs, which highlights an important security issue for the community, my main concern lies in the limited technical innovation compared to existing work. The contribution is incremental, primarily a modification to the chain-of-search which constrains the search space to a limited set of invisible variation selecors, and, in my opinion, does not meet the standard for ICLR.

2. The related work and experimental comparison could be enhanced by including and discussing recent attack methods, e.g.
[1] Paulus, Anselm, et al. "AdvPrompter: Fast Adaptive Adversarial Prompting for LLMs." Forty-second International Conference on Machine Learning (2025).
[2] Chen, Xuan, et al. "When llm meets drl: Advancing jailbreaking efficiency via drl-guided search." Advances in Neural Information Processing Systems 37 (2024): 26814-26845.

3. It is unclear how the proposed attack performs on defense methods, such as perplexity-based methods [3], paraphrase-based defenses [4], and adversarial-suffix-specific defenses [5].
[3] Alon, Gabriel, and Michael Kamfonas. "Detecting language model attacks with perplexity." arXiv preprint arXiv:2308.14132 (2023).
[4] Jain, Neel, et al. "Baseline defenses for adversarial attacks against aligned language models." arXiv preprint arXiv:2309.00614 (2023).
[5] Cao, Bochuan, et al. "Defending Against Alignment-Breaking Attacks via Robustly Aligned LLM." Proceedings of the 62nd Annual Meeting of the Association for Computational Linguistics (Volume 1: Long Papers). 2024.

**Questions:**

Please refer to the concerns highlighted in the Weaknesses.

---

### Official Review · Reviewer_6czL · 2025-11-01

**Soundness:** 2
**Presentation:** 3
**Contribution:** 2
**Rating:** 4
**Confidence:** 3

**Summary:**

This paper proposes an "imperceptible jailbreak" attack against Large Language Models. The core method utilizes a class of invisible Unicode characters called "Variation Selectors" by appending them as an adversarial suffix to malicious questions. Although these characters are not rendered on-screen, they are processed by the model's tokenizer, thereby altering the input token sequence. The authors design a "chain-of-search" pipeline to optimize these invisible suffixes to induce harmful responses. Experiments show this method achieves a high attack success rate on four open-source models.

**Strengths:**

1. The paper uncovers that a specific type of character, Unicode variation selectors, can be encoded by LLM tokenizers and subsequently influence the model's internal representations.

2. For this unique and constrained search space, the "chain-of-search" algorithm designed by the authors, which uses random search and bootstrapping, is a reasonable optimization strategy.

**Weaknesses:**

1. The paper's motivation is flawed. It focuses heavily on "imperceptible" attacks within the context of jailbreak. However, in a standard jailbreak threat model, the attacker controls the input and does not care if the adversarial prompt is visible. The "imperceptible" property is largely irrelevant to this task, though it would be critical for other attacks like copy-paste attack, which the paper does not make its primary focus.

2. Consequently, if "imperceptibility" is not a meaningful contribution to jailbreak, the contribution is reduced to finding a new character set, Unicode variation selectors, for building adversarial suffixes. This is a limited advance over existing methods, such as  GCG, that apply similar optimization techniques to a visible "vocabulary." Moreover, the method's performance does not consistently outperform traditional visible attacks.

3. This attack can be easily defended by removing the meaningless Unicode at the beginning, compared to GCG, which is hard to remove a meaningful word.

**Questions:**

1. Could the authors clarify why "imperceptibility" is an important or desirable feature for a jailbreak attacker? As argued in the weaknesses, this feature seems orthogonal to the primary goal of a jailbreak attack, which is simply to bypass safety alignment.
2. The paper ablates suffix length $L$ against the log-probability of the target token in Figure 7 . Is there a more direct trade-off analysis between the suffix length $L$ and the final ASR? What is the minimum length $L$ required to achieve a certain non-trivial ASR (e.g., 50% or 80%)?
3. There is a significant inconsistency in the baseline comparisons in Table 3. Llama-2-Chat-7B is compared against five different baselines, while Llama-3.1-Instruct-8B is compared against zero. Why were the visible SOTA baselines not evaluated against Llama-3.1-Instruct-8B?
4. For the prompt injection experiments in Section 4.4 (Table 4) , the method is only compared against weak baselines ("None" and "Random Variation Selectors"). Why were other state-of-the-art prompt injection methods not included as baselines to provide a more comprehensive comparison?

---

### Note · Authors · 2025-12-09

I have read and agree with the venue's withdrawal policy on behalf of myself and my co-authors.